# Imagination Helps Visual Reasoning, But Not Yet in Latent Space

**You Li** [1 2]   **Chi Chen** [3]   **Yanghao Li** [3]   **Fanhu Zeng** [3]   **Kaiyu Huang** [1 2]   **Jinan Xu** [1 2]   **Maosong Sun** [3]

## Abstract

Latent visual reasoning aims to mimic human's *imagination* process by meditating through hidden states of Multimodal Large Language Models. While recognized as a promising paradigm for visual reasoning, the underlying mechanisms driving its effectiveness remain unclear. Motivated to demystify the true source of its efficacy, we investigate the validity of latent reasoning using Causal Mediation Analysis. We model the process as a causal chain: the input as the treatment, the latent tokens as the mediator, and the final answer as the outcome. Our findings uncover two critical disconnections: (a) **Input-Latent Disconnect**: dramatic perturbations on the input result in negligible changes to the latent tokens, suggesting that latent tokens do not effectively attend to the input sequence. (b) **Latent-Answer Disconnect**: perturbations on the latent tokens yield minimal impact on the final answer, indicating the limited causal effect latent tokens imposing on the outcome. Furthermore, extensive probing analysis reveals that latent tokens encode limited visual information and exhibit high similarity. Consequently, we challenge the necessity of latent reasoning and propose a straightforward alternative named `CapImagine`, which teaches the model to explicitly *imagine* using text. Experiments on vision-centric benchmarks show that `CapImagine` significantly outperforms complex latent-space baselines, highlighting the superior potential of visual reasoning through explicit imagination. Our project is open-sourced at *here*.

---

[1]School of Computer Science and Technology, Beijing Jiaotong University [2]Key Laboratory of Big Data & Artificial Intelligence in Transportation (Beijing Jiaotong University), Ministry of Education [3]Tsinghua University. Correspondence to: Chi Chen <chenchithu@gmail.com>, Yanghao Li <goddy1027@gmail.com>.

*Proceedings of the $43^{rd}$ International Conference on Machine Learning*, Seoul, South Korea. PMLR 306, 2026. Copyright 2026 by the author(s).

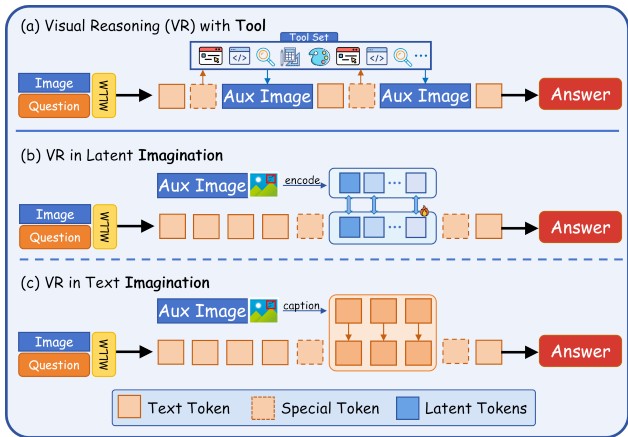

*Figure 1.* Comparison between visual reasoning with tools and through imagination. (a) Reasoing with tools perceive visual content through function calling such as zoom-in or drawing. (b) Latent-space imagination exploits the hidden states of MLLMs to conduct visual reasoning. (c) We show that imagination can be more effective in text-space.

## 1. Introduction

The field of Visual Reasoning within Multimodal Large Language Models (MLLMs) has witnessed a surge in interest, driven by steady progress in complex mathematical reasoning, spatial understanding, long-horizon planning, and fine-grained visual perception (Yang et al., 2025b; Bai et al., 2025c; Li et al., 2025e; Hong et al., 2025; Yang et al., 2025a; Ding et al., 2026; Lou et al., 2025; Zeng et al., 2026). The increasing complexity of visual reasoning tasks requires MLLM to employ more active percpetion of visual content. To meet this demand, visual reasoning with tools (Zhang et al., 2025c; Hong et al., 2025; Zhao et al., 2025b;a) generates interleaved multimodal reasoning trajectories, effectively incorporating visual semantics during the reasoning process. However, the obtained auxiliary image during reasoning process mostly comes from limited set of rigid tools, posing huge gap with human-like native imagination.

To address this gap, Latent Visual Reasoning (LVR) (Yang et al., 2025c) emerges as a novel paradigm that reasons through the hidden state in MLLMs, which we refers as latent token. Since input embeddings from different modalities have been aligned within the model, LVR trains MLLMs to output latent tokens that are compatible with visual embeddings and encode rich visual semantics. By deliberating

in this high-dimensional latent space, LVR enables a broader and less constrained form of **Visual Imagination** (Atwood, 1971; Pylyshyn, 2002). Building on these properties, a series of latent-space visual reasoning methods have been proposed (Li et al., 2025b; Wang et al., 2025b; Tong et al., 2025; Li et al., 2025b; Liu et al., 2025; Wang et al., 2026; Zhang et al., 2025a;b). These approches typically supervise latent tokens using visual features or hidden representations from the teacher model. Empirically, they exhibit strong performance across various vision-centric tasks.

Despite these promising results, the internal mechanism of Latent Visual Reasoning and the behavior of latent tokens are still poorly understood. In particular, it is unclear whether and how MLLM actually performs deliberative reasoning within the latent space. To address this gap, we adopt Causal Mediation Analysis framework and conceptualize latent reasoning as a causal process from input $X$ to intermediate latent tokens $Z$, and finally to output $Y$, i.e., $X \rightarrow Z \rightarrow Y$. Our analysis focuses on systematic perturbations on both $X$ and $Z$ to examine the full causality.

We begin by conducting instance-level perturbations on the input $X$, where the entire input sequence is altered. Surprisingly, the resulting latent tokens $Z$ exhibit a high degree of homogeneity measured by cosine similarity, even across diverse inputs and tasks. This similarity indicates a disconnect in the $X \rightarrow Z$ causality. Taken a step further, we implement systemtic intervention analysis and probing analysis on $Z$. Across multiple latent reasoning methods and diverse benchmarks, we find that drastic perturbations to $Z$ lead to negligible changes in final answer. Moreover, probing analysis reveals that latent tokens encode only minimal task-relevant visual semantics and are insufficient to support downstream reasoning on their own. The intervention and probing analysis demonstrates a disconnect in the $Z \rightarrow Y$ causality Collectively, these results show that latent tokens neither vary meaningfully according to the inputs, nor do they actually affect the final answer.

These observations naturally leads to a second question: how can MLLMs perform visual reasoning? To explore this, we propose a simple yet effective method under a strictly controlled setting. Specifically, we convert the original Monet-SFT-125K (Wang et al., 2025b) training data into a text-space imagination format. For each interleaved reasoning image, we generate textual descriptions of visual manipulations, such as highlighting or zooming into regions of interest, and train the MLLM to internalize these operations purely through text. Using the same data source as Monet, this simple data reformulation strategy yields substantially stronger results than latent-space approaches. Extensive evaluations on V* (Wu & Xie, 2024), HR-Bench (Wang et al., 2025c), MME-RealWorld-Lite (Zhang et al., 2024), and other vision-centric benchmarks show consistent im-

provements, surpassing Monet by 4.0% on HR-Bench-8K and 4.9% on MME-RealWorld-Lite. We believe our work sheds light on how to build more faithful, interpretable, and causally effective visual reasoning methods.

In conclusion, our contribution can be concluded as follows:

- We conduct a systematic study on latent tokens in visual reasoning through Causal Mediation Analysis, revealing that latent tokens contributes little to the causal reasoning process.

- We propose a simple yet effective text-space imagination method `CapImagine`, showing better causality than latent-space methods.

- Our method substantially outperforms latent-space approaches across multiple vision-centric benchmarks, demonstrating strong effectiveness and generality.

## 2. Related Work

### 2.1. Visual Reasoning with Tools

Tool-augmented visual reasoning approaches actively engage with the visual modality, adaptively perceiving visual content through explicit manipulation to lead to the final answer. These methods can be further distinguished by how intermediate visual observations are produced. Some works rely on fixed tool set such as zoom-in or image-drawing operations (Zheng et al., 2025; Qi et al., 2024; Lai et al., 2025; Jiang et al., 2025; Cao et al., 2025; Fu et al., 2025; Chen et al., 2025) to actively perceive the visual elements, dramatically expanding perceptual bandwidth compared with static perception. From the cognition and knowledge perspective, another stream of work (Wu et al., 2025a; Yu et al., 2026; Narayan et al., 2025) seeks to utilize retrieval or web-search tools for factual verification and external multimodal knowledge injection. Expanding the scope of predefined tool set, other approaches leverage self-rendered code to enable more flexible and free-form visual manipulations (Zhao et al., 2025b; Geng et al., 2025; Hong et al., 2025), faciliating agentic MLLM in visual reasoning.

### 2.2. Visual Reasoning through Imagination

Visual Imagination could be achieved through self-generation or latent-space reasoning. Unified multimodal models attempt to visually imagine through its inner generation ability, explicitly instantiating internal reasoning states (Deng et al., 2025; Li et al., 2025c; Shi et al., 2025). Latent visual reasoning proposes to conduct imagination through the hidden states in MLLMs, without decoding it into specific text token. Latent visual reasoning was first introduced by Mirage (Yang et al., 2025c), which addresses the challenge of latent supervision design by compressing

visual features extracted from intermediate reasoning images. Subsequent works (Li et al., 2025b; Tong et al., 2025; Dong et al., 2025; Zhang et al., 2025a) largely follow adopting vision encoder features as supervision signals, and further extend latent reasoning to broader perception scenarios, more flexible latent formats, and improved strategies for selecting supervisory visual features.

However, visual features are inherently continuous and semantically sparse. The compression strategy in Mirage tends to dilute discriminative semantics, and directly supervising latents with entire visual token sequences (Li et al., 2025b) can lead to latent mode collapse during inference. Monet (Wang et al., 2025b) introduces a distillation-based framework (Shen et al., 2025) that restricts gradient propagation exclusively to latent tokens, thereby preserving informative semantics from both intermediate images and key textual cues. Despite these advances, the LVR field still lacks rigorous investigation of many core design choices and mechanisms, an issue this paper aims to address.

# 3. Analysis: Latent Tokens Hardly Helps

## 3.1. Formulation

Latent Visual Reasoning refers to a reasoning paradigm in which the last hidden states of the final transformer layer are treated as latent tokens for solving visual question answering tasks. Given a set of input images $I_i{}_{i=0}^{N}$ and a question $q$, the model is required to produce an answer conditioned on the joint input $X = (\{I_i\}_{i=0}^{N}, q)$. During inference, the model $\mathcal{M}$ can adaptively switch between decoding normal text tokens and latent tokens. The inference process is formally defined as:

$$h_i = \mathcal{M}\left(E(x); y_{<i}\right), \quad y_0 = \emptyset \qquad (1)$$

$$y_i = \mathbb{I}(i \in \mathcal{I}_L) \cdot \phi(h_i) + \mathbb{I}(i \notin \mathcal{I}_L) \cdot E\left(\text{Decode}(h_i)\right) \quad (2)$$

where $\mathcal{I}_L$ denotes the index set of latent tokens, $\phi(h_i)$ is an optional projection layer applied to hidden states, and $E(\cdot)$ represents the embedding process. The indicator function $\mathbb{I}(\cdot)$ determines whether the current decoding step operates in latent mode or normal text mode.

In practice, the number of latent tokens is usually predefined. The latent mode starts immediately after the model outputs <|latent_start|>. During latent mode, the model takes the last hidden state as the input for the next step. The model exits latent mode when the current hidden state is decoded as <|latent_end|>, after which normal text decoding resumes.

When text and latent tokens are interleaved together, our primary research focus are the latent tokens, denoted as $Z$. Given the original input $X$ and the whole reasoning process, the model ultimately produces the final answer $Y$. Explicitly tracing the role of latent tokens in visual reasoning, we abstract the overall reasoning process as:

$$X \to Z \to Y \qquad (3)$$

In the following content, we will conduct targeted interventions $P(Z \mid do(X))$ and $P(Y \mid do(Z))$ to elucidate the role of latent tokens in visual reasoning.

## 3.2. Causal Analysis of $X \to Z$

> **Finding 1:** Latent tokens are similar across instances and tasks, and progressively collapse into highly identical states.

We begin with a causal mediation analysis (Pearl, 2009) conducting instance-level perturbations on the input $X$ and measure how the latent reasoning tokens change accodingly with the entire input sequence altering, i.e., $P(Z \mid do(x))$.

**Experiment Setting** We evaluate three representative baselines: (1) *Monet* (Wang et al., 2025b), a distillation-based model focused on general scenarios; (2) *LVR* (Li et al., 2025b), which leverages image features as supervision in general scenarios; and (3) *Mirage* (Yang et al., 2025c), which also uses image features but is fine-tuned for task-specific settings. For general VQA scenarios, we uniformly sample instances from V* (Wu & Xie, 2024), MME (Yin et al., 2024), OCRBench-v2 (Fu et al., 2024a), MME-Realworld-Lite(Zhang et al., 2024), and TableVQA (Kim et al., 2024), resulting in a total of 100 testing instances. These instances have been sorted and grouped in results visualization. For the task-specific Mirage model, we adopt its released Visual Spatial Planning dataset, which requires the model to reach the destination circumventing obstacles such as frozen lakes.

During inference, all three models are prompted to perform latent reasoning and only instances with valid latent reasoning process are reserved. As illustrated in Figure 2, we examine latent tokens from two perspectives: *inter-instance*, by sampling latent tokens at fixed positions across different instances; *intra-instance*, by sampling all latent tokens within a single instance. The intra-instance results are then averaged across all instances. Additionally, we have considered the pattern of text tokens, image tokens and the inner representation of MLLM after the input sequence. For intra-instance pattern of textual reasoning, we analyze the hidden states of first 16 tokens during generation.

**Inter-instance Analysis.** As shown in Figure 2, latent tokens at the same position across different instances exhibit consistently high cosine similarity. This indicates that these latent tokens encode little information from the input images or questions. Additionally, latent tokens from different tasks also remain highly similar, suggesting that the they also fail to capture coarse task-level distinctions. Furthermore, the degree of similarity intensifies as the reasoning continues,

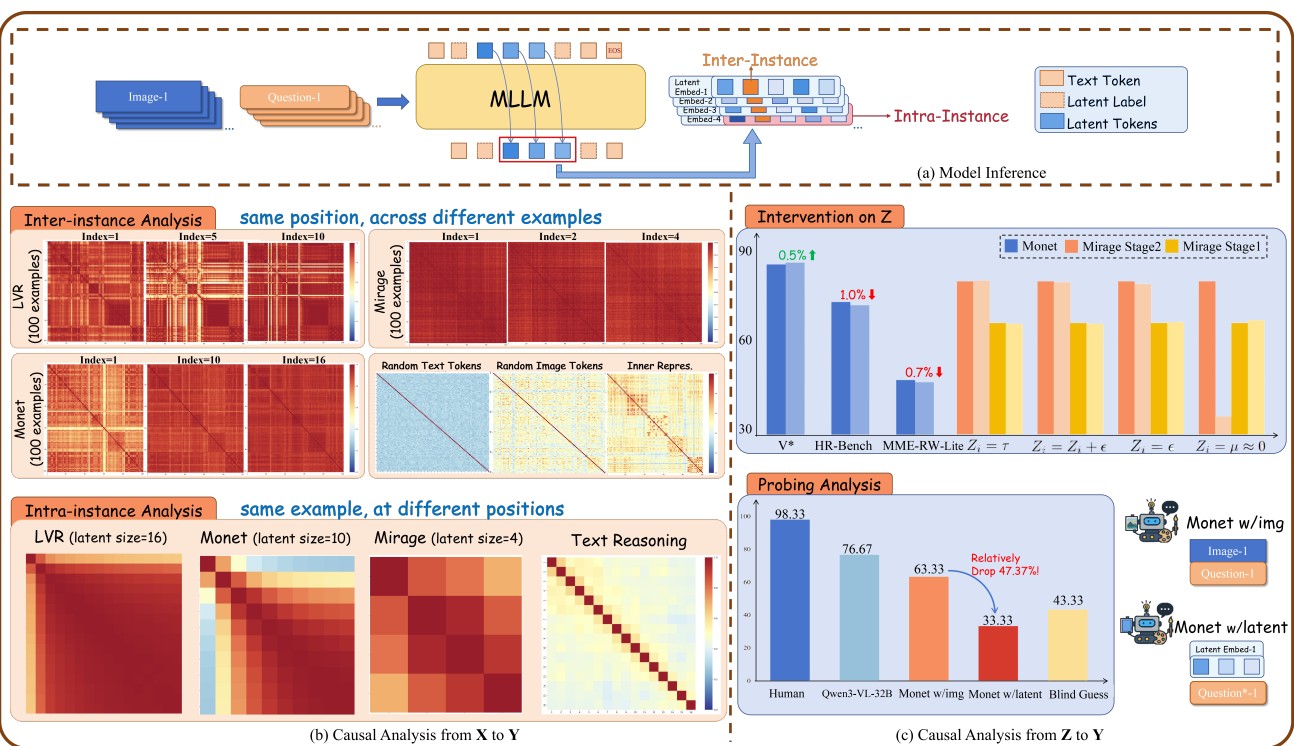

*Figure 2.* **Our systematic latent analysis framework** for investigating the internal mechanisms and behavioral patterns of latent tokens. (a) Model Inference illustrates the latent generation procedure and the derivation of latent sequences. (b) and (c) delineate the causal analysis pipelines for $X \rightarrow Z$ and $Z \rightarrow Y$, respectively. The former encompasses **Inter-instance and Intra-instance Analysis**, whereas the latter comprises **Intervention Analysis and Probing Analysis**. In diagram Intervention on $Z$, $\tau$ denotes a fixed tensor, $\epsilon$ represents random Gaussian noise with $\epsilon \sim \mathcal{N}(0, \sigma^2)$, and $\mu$ is a small value close to zero.

with all latent tokens increasingly degenerating as more latent tokens are generated. In contrast, text/image tokens and inner representation of MLLM all carry informative and distinctive semantics, exhibiting low similarity cross instances and tasks.

**Intra-instance Analysis.** When examining the average latent token behaviour within individual instances under input $X$ alteration, all three models display a progressive degeneration phenomenon: latent tokens collapse into clusters of highly similar representations as reasoning continues. With more autoregressive steps, the LLM backbone applies increasingly smaller modification to the latent states, causing tokens to converge toward uniform representations. Specifically, the LVR model degenerates the fastest, with token collapse occurring as early as at the second step. Monet initially produces semantically rich latent tokens, but they gradually lose distinctiveness by the fifth step. In comparison, the hidden states similarity of text reasoning are dramatically lower. This reveals the clear and steady states transition in text generation and obscure reasoning trajectory in latent reasoning.

**Latent Pattern across Various Models.** Across different approaches, distinct latent patterns emerge. As shown in

the Inter-instance Analysis in Figure 2 (b) and Appendix B, Monet exhibits slower degeneration speed at different latent index, but ultimately converges into a highly uniform latent space. As for LVR, although it collapses rapidly, some latent tokens retain partial distinctiveness even after lengthy reasoning. By contrast, Mirage, which compresses the original lengthy visual tokens into a few latent tokens, demonstrates minimal distinctiveness throughout the entire reasoning process.

### 3.3. Causal Analysis of $Z \rightarrow Y$

Despite the degenerate nature of the latent tokens, they may still helpful to get the final answers. To investigate, we first directly intervating on the latent tokens $Z$ to explicitly diagnose its causal effect on the final answer $Y$, then conduct a probing analysis to test whether $Z$ sufficiently lead to $Y$.

#### 3.3.1. INTERVENTION ON LATENT TOKENS $Z$

> **Finding 2:** **Fundamental change on latent tokens $Z$ only results in minimal change on answers $Y$.**

**Experiment Setting.** We conduct interventions $do(Z)$ on both *Monet* and *Mirage*, representing general-purpose and

task-specific scenarios. For Monet, we apply a strong intervention by forcing all latent tokens across different positions and instances as an shared identical tensor. For Mirage, we further explore more diverse intervention strategies. Besides the intervation strategy on Monet, we also consider (1) injecting Gaussian noise into the latent tokens, (2) replacing latent tokens entirely with Gaussian noise, and (3) setting all latent tokens to a small value close to zero. To preclude out-of-distribution shifts, the generated noise is explicitly parameterized to match the empirical mean and standard deviation of the original latent tokens.

**Results Analysis.** The experimental results are summarized in Figure 2 (c) Intervention on $Z$ and detailed in Appendix B Table-4. Surprisingly, across V*, HR-Bench, and MME-Realworld-Lite, these drastic alteration applied to the latent tokens $Z$ result in only marginal answer variations. On V*, overall performance even exhibits a slight improvement by 0.5%. Only minor degradation is observed on the HR-Bench-4K and on MME-RealWorld-Lite, by 1.0% and 0.7% respectively. Overall, even fundamental alterations to the latent tokens lead to negligible performance fluctuations, suggesting that these latent tokens $Z$ exerts limited influence on the final output.

We further evaluate Mirage on the VSP dataset (Yang et al., 2025c; Wu et al., 2025b), considering both its stage-1 and stage-2 variants. Dramatic decline only happens when setting latent tokens as small value on stage-2 variant, where the intervention is so strong and results in repetition. In other cases, even fundamental change of latent tokens such as directly replacing the latent tokens with Gaussian noise result in negligible changes. These findings consistently indicate that the model neither meaningfully attends to the latent tokens nor encodes critical information within them.

### 3.3.2. PROBING ANALYSIS ON LATENT TOKENS $Z$

> **Finding 3:** Latent tokens encode limited visual semantics and are insufficient for accurate answer derivation.

**Experiement Setting.** Here, we further diagnose the causal effect $Z$ imposing on answer $Y$ by a probing analysis on latent tokens $Z$. In this analysis, we focus on *Monet*, whose latent supervision is obtained by jointly optimizing over visual signals and textual semantics in the original interleaved multimodal reasoning data. Through multi-stage training, these latent tokens are encouraged to encode informative visual semantics that facilitate solving the question.

Specifically, we sample question–image pairs $\{(I_i, q_i)\}_{i=0}^N$ from V* and collect the corresponding latent embeddings $\{Z_i\}_{i=0}^N$ generated during inference. These embeddings are expected to encode key visual evidence supporting not only the initial task but also other related queries grounded in the same visual content. To examine the semantics captured by

the latent tokens, we further construct 30 multiple-choice VQA questions $\{(Z_i, \tilde{q}_i)\}_{i=0}^N$ that focus on the same image regions but probe different attributes of the referenced objects. If the latent tokens effectively capture essential visual semantics, they should support solving these derived questions. Example are provided in the Appendix A.

**Result Analysis.** The experimental results are summarized in Figure 2 (c). The results reveal that directly using latent tokens as the sole input leads to notably weak performance, falling behind even text-only guessing baselines. In contrast, when the original image is provided, both Monet and Qwen3-VL-32B (Bai et al., 2025a) achieve strong performance, reaching 76.67% accuracy, which also validates the quality and consistency of our manually curated questions. Taken together, these findings cast huge doubt on the extent to which the latent tokens alone effectively capture and preserve actionable visual semantics within the model's reasoning process.

### 3.4. Summary

In summary, these highly homogeneous latent tokens (Findings-1) contribute marginally to the final prediction. The model potentially adopts an implicit shortcut circumventing the latent visual reasoning pathway (Findings-2). Moreover, the encoded semantics of latent tokens are also minimal (Findings-3). So far, the full potential of latent tokens in current methods **has not yet been fully discovered**, and the latent tokens are behaving similarly with soft prompt or placeholders instead of active carrier of visual imagination or reasoning.

## 4. CapImagine

### 4.1. Method Design

The essence of visual imagination primarily lies in *interleaved multimodal reasoning*, where internal visual thought could be explicitly outlined and evolve alongside the textual reasoning chain. Existing Latent Visual Reasoning methods attempt to internalize such visual thoughts into latent tokens. However, as shown in the previous sections, these latent representations fail to preserve meaningful visual semantics and contribute little to downstream reasoning.

Motivated by this limitation, we explore whether text-space reasoning can more effectively retain the essential information embedded in interleaved data and support visual imagination. Instead of relying on latent variables, we convert the semantic changes introduced by intermediate images into textual captions. This forces the model to imagine visual transformations over the original image through an explicit text-space reasoning chain. Unlike prior text-space reasoning approaches, our method is grounded in concrete intermediate visual evidence. By verbalizing visual transitions that

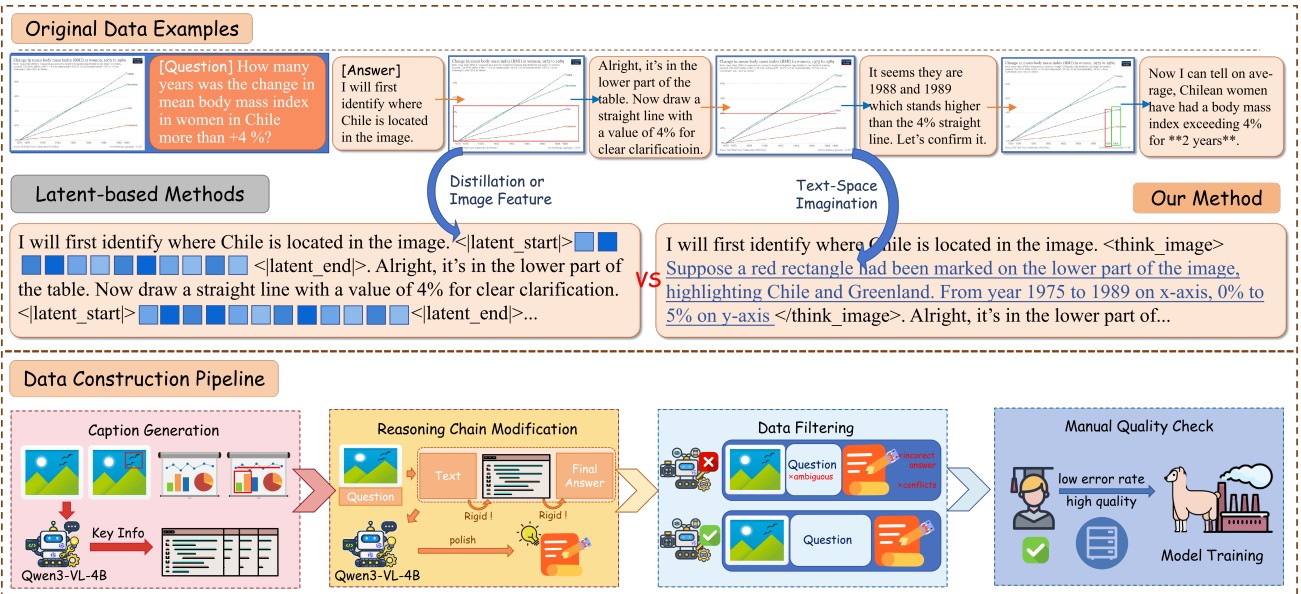

*Figure 3.* **Illustration of Our Method and Data Construction Pipeline**, through which we conduct a strictly controlled training setting with Monet for fair and convincing comparisons. The upper section presents the interleaved format of original data. The middle section clarifies the key methodological differences between the two approaches. The lower section shows the data construction procedures.

would otherwise occur in hidden space, the model performs imagination as if intermediate images were present.

### 4.2. Dataset Construction

**Data Rewriting.** Specifically, our data construction is based on the Monet-SFT-125K (Wang et al., 2025b) dataset and adopts two forms of image rewriting. For the Visual-CoT and Zebra-CoT visual search (Li et al., 2025a) subsets, which primarily focus on zooming into key image regions, we provide the original question together with the highlighted image region to Qwen3-VL-4B (Bai et al., 2025a), prompting it to generate concise and accurate captions that refocus the highlighted visual semantics. For other subsets such as Refocus (Fu et al., 2025) and CogCoM (Qi et al., 2024), which involve direct image manipulations such as marking or drawing auxiliary lines, we present both the original image and its manipulated counterpart to Qwen3-VL-4B. The model is instructed to describe the visual differences and explicitly verbalize the key information revealed by the manipulation, such as marked numerical values or highlighted textual entities. Through this process, language fully carries the semantics of auxiliary images, effectively bypassing latent representations.

However, directly inserting the rewritten text into the original reasoning trajectories often results in rigid transitions and disrupts logical coherence. To address this issue, we further employ the MLLM to globally refine the reasoning chains. This step corrects potential inconsistencies and improves fluency, allowing the newly generated textual descriptions to integrate smoothly into the original reasoning

process. The above data construction pipeline is in Figure 3.

**Data Filtering.** Although the Monet-SFT-125K dataset has already undergone rigorous filtering procedure, the inherently low quality of the Visual-CoT (Shao et al., 2024) data, which accounts for 94.88% of Monet-SFT-125K, significantly undermines the effectiveness of our data rewriting strategy. We identify two primary issues. First, the final answer of the original question often conflicts with the newly generated visual observations, resulting in misalignment between the reasoning process and the answer. Second, a large portion of questions in the Visual-CoT subset are overly ambiguous or fundamentally unanswerable, lacking a clear reference to the target object. As a result, early experiments using the raw rewritten data yield only limited improvements on downstream tasks.

To mitigate these issues, we perform a comprehensive quality assessment of each training instance through MLLM. The evaluation focuses on both the correctness of the reasoning process and the degree of question ambiguity, and instances with evident flaws are filtered out. Manual inspection confirms the effectiveness of this automated filtering procedure. After filtering, we retain 17k high-quality training instances. To eliminate the effect of data quantity difference with Monet-SFT-125K, we conduct strict ablation study in Section 5.3, ensuring a fair comparison with Monet.

*Table 1.* **Performance comparison across perception-centric and visual reasoning benchmarks**, where our method consistently outperforms competing baselines. The best results are highlighted in **bold**. Results marked with '*' are reported from prior work.

| Model | V* | | | HRBench4K | | | HRBench8K | | | MME-RealWorld-Lite | | | BLINK | |
|---|---|---|---|---|---|---|---|---|---|---|---|---|---|---|
| | Overall | Attr. | Spa. | Overall | FSP | FCP | Overall | FSP | FCP | Overall | Rea. | Perc. | Jigsaw | MV. |
| **Proprietary Model** | | | | | | | | | | | | | | |
| GPT-4o | 67.5* | 72.2* | 60.5* | 59.0* | 70.0* | 48.0* | 55.5* | 62.0* | 49.0* | 52.0* | 48.3* | 54.4* | 55.3* | 59.4* |
| **Open-Source Models** | | | | | | | | | | | | | | |
| InternVL3-8B | 72.3 | 73.0 | 71.1 | 70.8 | 79.3 | 62.3 | 62.0 | 64.3 | 59.8 | 47.9 | 42.9 | 51.1 | 50.0 | 45.9 |
| Qwen2.5VL-7B | 76.4 | 77.4 | 75.0 | 68.0 | 80.3 | 55.8 | 63.8 | 73.8 | 53.8 | 45.8 | 39.7 | 49.6 | 62.7 | 42.9 |
| **Reasoning with Tool Methods** | | | | | | | | | | | | | | |
| PixelReasoner | 80.6 | 83.5 | 76.3 | 72.9 | 86.0 | 60.3 | 66.9 | 80.0 | 54.3 | 49.7 | 44.5 | 53.1 | - | - |
| DeepEyes | 90.0 | 92.1 | 86.8 | 75.1 | 91.3 | 59.0 | 72.6 | 86.8 | 58.5 | 53.2 | 45.6 | 58.1 | - | - |
| **Reasoning through Imagination Methods** | | | | | | | | | | | | | | |
| LVR | 81.7 | 84.4 | 77.6 | 70.8 | 83.8 | 57.8 | 63.0 | 74.5 | 51.5 | 50.6 | 42.7 | 55.7 | 52.0 | 46.6 |
| Monet | 83.3* | 83.5* | 82.9* | 71.0* | 85.3* | 56.8* | 68.0* | 79.8* | 56.3* | 46.9 | 40.3 | 51.2 | 50.0 | 47.4 |
| + w/ subset | 79.6 | 82.6 | 75.0 | 70.7 | 88.0 | 53.3 | 67.9 | 84.0 | 51.8 | - | - | - | - | - |
| CapImagine | **85.9** | **87.8** | **82.9** | **74.1** | 88.5 | **59.8** | 70.7 | 84.8 | 56.5 | 54.8 | 48.5 | 58.9 | 64.7 | 49.6 |
| + w/o Rewriting | 82.7 | 87.8 | 77.6 | 74.1 | **89.0** | 58.3 | 69.8 | 83.5 | 56.0 | 53.5 | 43.7 | 57.6 | 59.3 | 42.9 |
| + w/o Filtering | 82.7 | 82.6 | 82.9 | 72.5 | 88.3 | 56.8 | 69.3 | 81.8 | 56.8 | 46.1 | 40.9 | 49.5 | 56.0 | 44.4 |

# 5. Experiments

## 5.1. Experiment Setup

**Benchmarks.** To comprehensively assess the effectiveness of text-driven visual imagination, we adopt a diverse set of high-resolution (HR) visual perception benchmarks following the experimental protocol of Monet: V*, HR-Bench-4K, HR-Bench-8K, and MME-RealWorld-Lite, which emphasize fine-grained perception under high-resolution settings. Beyond the zoom-in ability, we further incorporate subsets of BLINK (Fu et al., 2024b), Hyperphantasia (Sepehri et al., 2025) and STARE (Li et al., 2025d) rigorously evaluate advanced visual reasoning capabilities within highly abstract contexts. Finally, we evaluate on TableVQA to examine the generalization of our approach in diagram and table images.

**Baselines.** We compare against three categories of baselines. (1) Open-source models: we evaluate InternVL3-8B (Zhu et al., 2025) and Qwen2.5-VL-7B (Bai et al., 2025b), the latter serving as the backbone model for all following baselines. (2) Reasoning with Tool methods, including DeepEyes and PixelReasoner (Wang et al., 2025a; Zhao et al., 2026), which leverage reinforcement learning to enhance perception via zoom-in operations. (3) Reasoning through Imagination Methods, namely LVR and Monet, which perform visual imagination in latent space across general scenarios. (4) Text-space reasoning methods, including PAPO (Wang et al., 2025d), Vision-R1 (Huang et al., 2025), R1-Onevision (Yang et al., 2025b), which conduct long-chain reasoning directly with explicit text. We provide the results for these baselines in Appendix section C. Additionally, we also report results from the proprietary GPT-4o (Hurst et al., 2024) model. For Monet, we adopt an LLM-as-a-judge protocol to extract final answers, aligning with prior practice.

**Training Details.** Our model is built upon Qwen2.5-VL-7B and trained on reconstructed data from Monet-SFT-125K. We perform CoT-SFT fine-tuning using the Monet codebase on 8 A800-80G GPUs, with a batch size of 1 and gradient accumulation of 16. To mitigate training instability and incentivate the full potential of the data, we select the best-performing checkpoint during training (Nishida et al., 2025).

## 5.2. Main Results

Across evalution on various perception-centric benchmarks, our method consistently outperforms the strong baseline Monet, achieving average improvements of 3.44% on HR-Bench and 2.6% on V*. On MME-RealWorld-Lite, our reproduced Monet shows only marginal gains over its base model, whereas our approach effectively handles diverse real-world queries. These results highlight the effectiveness of zoom-in-based visual imagination for fine-grained visual perception. Compared with reasoning with tools approaches, our method substantially outperforms PixelReasoner, while remaining slightly behind DeepEyes, suggesting that direct image replay does provide complementary benefits.

Beyond high-resolution perception, we further evaluate more abstract visual reasoning tasks. Jigsaw and multi-view reasoning require reconstructing global structure and performing spatial reasoning across views. Our method generalizes well to these settings, surpassing both LVR and Monet by over 10 points. On TableVQA, which emphasizes identifying and comparing key values, our approach achieves a 6.1% improvement over Monet. We provide the results of Hyperphantasia and STARE in the Apppendix C.

Overall, these results demonstrate that text-driven visual

*Table 2.* **Results on TableVQA benchmark.** Through imagination in text-space, our method consistently outperforms baselines.

| Model | TableVQA | | | |
|---|---|---|---|---|
| | VWTQ | VWTQ$_{syn}$ | VTabFact | Overall |
| Gemini-Pro 1.5 | 38.5 | 43.2 | 75.6 | 52.4 |
| LLaVA-NeXT-34B | 36.4 | 38.0 | 71.2 | 48.5 |
| VisProg | 53.2 | 62.0 | 76.4 | 63.9 |
| Phi-3-Vision | 44.7 | 53.2 | 74.4 | 57.4 |
| Monet | 55.3 | 60.4 | 78.8 | 64.8 |
| CapImagine | **60.9** | **68.0** | **83.2** | **70.7** |

imagination offers an effective mechanism for both fine-grained perception and abstract visual reasoning. Imagination does help visual reasoning, but not yet in latent space.

## 5.3. Ablation Study

We conduct controlled ablation studies to disentangle the effects of data rewriting, data filtering and methodology selection in the following ablation study.

**Data Rewriting.** To assess the role of data rewriting, we replace the text-space imagination descriptions in our training data with a single <think_image> token and fine-tune the model under identical settings. As shown in Table 1 (denoted by + w/o Rewriting), this modification leads to consistent performance degradation across all benchmarks, including a 3.13% drop on V*, confirming the effectiveness of text-driven visual imagination.

**Data Filtering.** We further examine the impact of data filtering by fine-tuning directly on the original Monet-SFT-125K dataset. To eliminate the training-inference misalignment in Monet, where auxiliary images are present during training but unavailable at inference, we replace intermediate images with the <think_image> token during supervised fine-tuning (denoted by + w/o Filtering). We find that training without data filtering results in another continual performance decline, demonstrating the necessity of quality control. Notably, after removing the training–inference mismatch, direct supervised fine-tuning on Monet-SFT-125K achieves performance comparable to Monet, which additionally undergoes a Policy Optimization stage. This observation further question the role of latent in visual imagination.

**Comparison under Data Parity.** Lastly, to establish strict data parity between CapImagine and Monet, we replicate the complete Monet training pipeline utilizing the identical 17K curated subset employed by CapImagine. As demonstrated in Table 1, training Monet on this restricted dataset (denoted as + w/ subset) yields only marginal performance improvements, substantially underperforming CapImagine and marginally trailing the full-data Monet baseline. This substantiates that, even under strictly controlled data conditions, explicit text-space reasoning remains more effective than current latent-space reasoning paradigms.

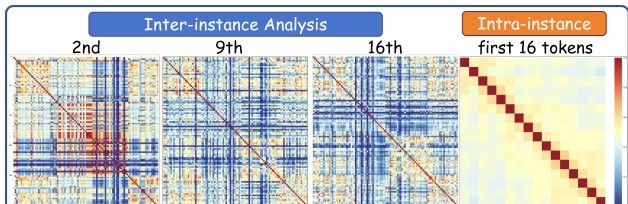

*Figure 4.* **Inter-instance and Intra-instance Analysis** of the inner hidden states of CapImagine during reasoning process.

*Table 3.* **Performance change under the intervetion** of the intermediate reasoning process of CapImagine.

| Model | V* | | | HR-Bench4K | | |
|---|---|---|---|---|---|---|
| | Avg | Attr. | Spa. | Avg | FSP | FCP |
| Qwen2.5-VL | 76.4 | 77.4 | 75.0 | 68.0 | 80.3 | 55.8 |
| CapImagine | 85.9 | 87.8 | 82.9 | 74.1 | 88.5 | 59.8 |
| CapImagine $do(Z)$ | 22.5 | 20.0 | 26.3 | 24.0 | 20.0 | 28.0 |
| $\Delta \downarrow$ | -63.4 | -67.8 | -56.6 | -50.1 | -68.5 | -31.8 |

## 5.4. Dependency Analysis

**Experiment Setup.** In this subsection, we conduct a causal mediation analysis on the text-form imagination variable $Z$, following perturbation protocols analogous to those in the previous section. For interventions on the input $X$, we perform instance-level modifications to the input sequence and analyze the resulting changes in the hidden states of text imagination tokens, considering both inter-instance and intra-instance similarities. For interventions on $Z$, we follow the protocol of (Zhang et al., 2025b) and explicitly manipulate the reasoning process. Specifically, CapImagine is first prompted to answer a question. The generated answer is then removed, and Qwen3-32B is used to deliberately alter the imagination content so that it leads to an incorrect conclusion. This corrupted reasoning trace is finally fed back to CapImagine, which is asked to complete the generation and produce the final answer.

**Results analysis.** As shown in Figure 4, inter-instance analysis yields consistently low cosine similarity, indicating a strong causal dependency between $X$ and $Z$. Intra-instance analysis further shows substantial diversity among consecutive hidden states, suggesting that each imagination token encodes distinct semantic content. Intervening on $Z$ produces a pronounced impact on the final prediction $Y$. When key information in the imagination content is modified, performance drops sharply below random-guess levels. Overall, from a causal mediation perspective, the text-form imagination process in CapImagine exhibits a substantially stronger and more direct causal influence than latent tokens, and plays a central role in the end-to-end reasoning process.

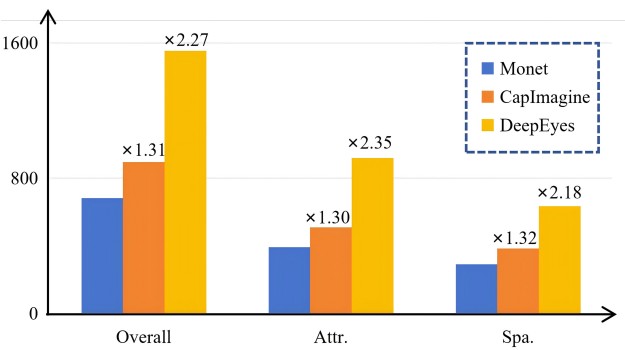

*Figure 5.* **Inference Speed Comparison** of Monet, CapImagine and DeepEyes on V*. The unit for Y-axis is Seconds (S). X-axis represents different categories of the V* benchmark.

### 5.5. Efficiency Analysis

Although `CapImagine` employs relatively long text-form imagination sequences, we compare its inference efficiency against the latent-space method Monet and the tool-augmented reasoning method DeepEyes. We measure only decoding time and ensure that all models generate complete answers. The results are summarized in Figure 5. `CapImagine` achieves inference speed comparable to Monet, despite operating entirely in text space. At the same time, it is nearly twice as fast as the reasoning with tools method DeepEyes while delivering competitive performance. These results demonstrate that `CapImagine` offers a favorable trade-off between effectiveness and efficiency, combining strong reasoning ability with practical inference cost.

## 6. Conclusion

In this work, we systematically investigate the internal mechanisms of latent-space visual reasoning methods through causal mediation analysis. Our results reveal that latent tokens are highly homogeneous, minimally sensitive to input, weakly result-oriented and semantically limited, failing to serve as effective carrier of visual imagination and genuine reasoning. To address this limitation, we propose a text-space imagination method exhibiting better causal effect and higher performance. We believe our study provides a rigorous investigation of current latent visual reasoning methods and offers guidance toward developing more faithful, interpretable, and effective latent reasoning approaches.

## Impact Statement

This paper presents work whose goal is to advance the field of machine learning. There are many potential societal consequences of our work, none of which we feel must be specifically highlighted here.

## Acknowledgement

The work is initiated and supported by AI9Stars Team.

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

## A. Examples for Derived Questions in Probing Analysis.

We present qualitative examples of derived questions used in our probing analysis. These questions focus on the same visual regions as the original queries while varying the queried attributes to examine the semantic consistency of latent representations. During probing analysis, the obtained latent embeddings and these derived questions are presented to the model for final answer generation.

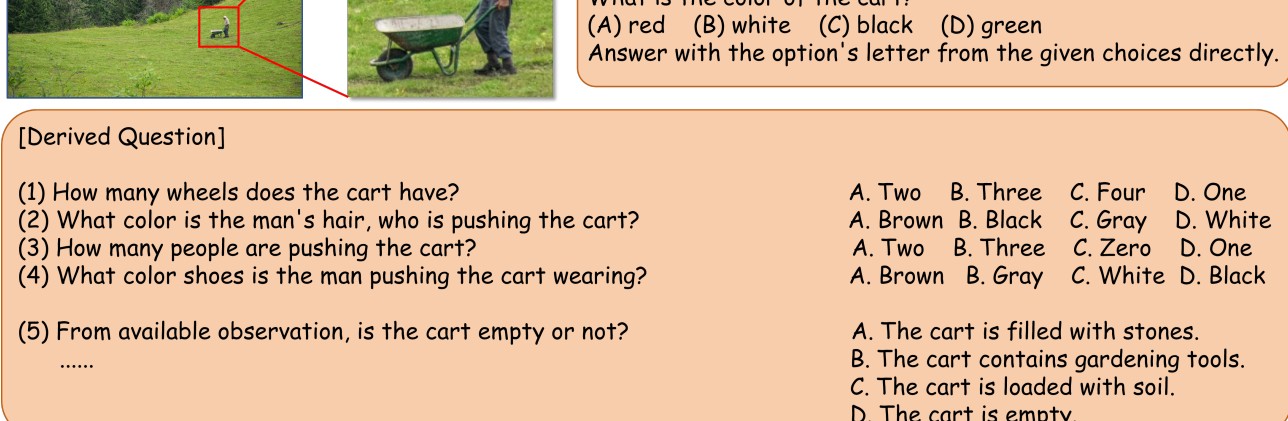

[Original Question]

What is the color of the cart?
(A) red    (B) white    (C) black    (D) green
Answer with the option's letter from the given choices directly.

[Derived Question]

(1) How many wheels does the cart have?                                A. Two    B. Three    C. Four    D. One
(2) What color is the man's hair, who is pushing the cart?             A. Brown  B. Black    C. Gray    D. White
(3) How many people are pushing the cart?                              A. Two    B. Three    C. Zero    D. One
(4) What color shoes is the man pushing the cart wearing?              A. Brown  B. Gray    C. White  D. Black

(5) From available observation, is the cart empty or not?              A. The cart is filled with stones.
    ......                                                             B. The cart contains gardening tools.
                                                                       C. The cart is loaded with soil.
                                                                       D. The cart is empty.

*Figure 6.* The derived questions focus on the same region or object as the original question, while differing in the queried attribute aspects.

## B. Detailed Results for Intervention on $Z$.

| Model | V* | | | HR-Bench-4K | | | MME-RealWorld-Lite | | |
|---|---|---|---|---|---|---|---|---|---|
| | Overall | Attribute | Spatial | Overall | FSP | FCP | Overall | Reasoning | Perception |
| Qwen2.5VL-7B | 76.4 | 77.4 | 75.0 | 68.0 | 80.3 | 55.8 | 45.8 | 39.7 | 49.6 |
| Monet | 82.7 | 84.4 | 80.3 | 71.1 | 87.3 | 55.0 | 46.9 | 40.3 | 51.2 |
| Monet $do(Z)$ | 83.3 | 85.2 | 80.3 | 70.1 | 87.3 | 53.0 | 46.2 | 39.9 | 50.3 |
| $\Delta$ | +0.5 | +0.9 | +0.0 | -1.0 | +0.0 | -2.0 | -0.7 | -0.4 | -0.9 |

| Model | $Z$ | $Z_i = \tau$ | $Z_i = Z_i + \epsilon \sim \mathcal{N}(0, \sigma^2)$ | $Z_i = \epsilon \sim \mathcal{N}(0, \sigma^2)$ | $Z_i = \mu \approx 0$ |
|---|---|---|---|---|---|
| Mirage-Stage1 | 64.2 | 64.0 | 64.0 | 64.5 | 65.0 |
| $\Delta$ | – | -0.2 | -0.2 | +0.3 | +0.8 |
| Mirage-Stage2 | 77.0 | 77.2 | 76.7 | 76.2 | 35.5 |
| $\Delta$ | – | +0.2 | -0.3 | -0.8 | -41.5 |

*Table 4.* **Performance variation under different latent interventions.** The upper table reports the results of Monet when all latent tokens are set to the same tensor, denoted by $do(Z)$. The lower table focuses on Monet under various latent intervention strategies. $\tau$ denotes a fixed tensor, $\epsilon$ represents random Gaussian noise with $\epsilon \sim \mathcal{N}(0, \sigma^2)$, and $\mu$ is a small constant close to zero. $Z_i$ denotes the latent token at the $i_{th}$ position.

## C. More Evaluation Results

We provide more comprehensive evaluation results in this section.

To compare `CapImagine` against other text-space reasoning methods, we have evaluated against a suite of strong recent baselines, including PAPO, Vision-R1, R1-OneVision, and LLaVA-OneVision, across diverse benchmarks (V*, HR-Bench, MME-Realworld-Lite, and BLINK). As shown in the Table 5, `CapImagine` consistently exhibits superior performance over strong baselines. This robust performance across multiple datasets thoroughly validates the efficacy and necessity of our proposed text-space imagination framework.

*Table 5.* Evaluation Results of Text-Space reasoning baselines on various benchmarks. Best results have been **bolded**.

| | V* | HR-Bench (4K&8K) | MME-RealWorld-Lite | BLINK |
|---|---|---|---|---|
| **Text-Space Reasoning Methods** | | | | |
| PAPO | 36.1 | 68.1 | — | 52.7 |
| Vision-R1 | 80.1 | 60.9 | 47.9 | 51.0 |
| R1-Onevision | 84.2 | — | 35.1 | 50.1 |
| LLaVa-OneVision | 75.4 | 61.4 | 48.5 | 53.0 |
| CapImagine | **85.9** | **72.4** | **54.8** | **54.8** |

For complex visual reasoning abilities, we additionally evaluate `CapImagine` on STARE and Hyperphantasia, targeting more imagination-intensive and spatial reasoning settings. The experimental results reveal a consistent performance advantage for `CapImagine` over the Monet baseline, highlighting the generalizability of our text-space imagination method not only in language-friendly tasks but more complex scenarios such as spatial perception and counting.

*Table 6.* Model Performance Without Visual Simulation across tasks in STARE.

| | 2D Trans. | 3D Trans. | Cube Net | Tangram | Temporal | Perspective | Overall |
|---|---|---|---|---|---|---|---|
| GPT-4o | 71.2 | 65.5 | 50.3 | 52.5 | 39.0 | 38.7 | 53.9 |
| Claude-3.5 Sonnet | 65.9 | 51.5 | 52.3 | 59.0 | 54.0 | 26.1 | 53.1 |
| Gemini-2.0 Flash | 69.5 | 56.1 | 37.7 | 65.0 | 38.6 | 37.2 | 51.3 |
| Qwen2.5VL-7B | 38.4 | 28.8 | 40.7 | 54.5 | 36.5 | 23.2 | 36.7 |
| Monet | 32.4 | 26.5 | 55.4 | 48.1 | 32.1 | 27.2 | 35.3 |
| CapImagine | 43.2 | 35.8 | 56.5 | 49.1 | 35.9 | 25.2 | 40.7 |

*Table 7.* Accuracy (%) of models on Hyperphantasia. We report the best accuracy for each puzzle across difficulties in **bold** and underscore the second-best accuracy.

| | Interpolation | | | | | | Extrapolation | | | | | | Mean | | |
|---|---|---|---|---|---|---|---|---|---|---|---|---|---|---|---|
| | Seven Segments | | | Connect the Dots | | | Linear Trajectory | | | Parabolic Trajectory | | | Mean | | |
| Model | Easy | Medium | Hard | Easy | Medium | Hard | Easy | Medium | Hard | Easy | Medium | Hard | Easy | Medium | Hard |
| Gemini 3 pro preview | **98** | **96** | **96** | **99** | **86** | 68 | 42 | **38** | **43** | 30 | **32** | 24 | 67.25 | 63.00 | 57.75 |
| o4-mini | 83 | 85 | 85 | 90 | 69 | 64 | 43 | 26 | 23 | 35 | 25 | **33** | 62.75 | 51.25 | 51.25 |
| GPT4-o | 3 | 4 | 0 | 96 | 80 | 59 | 28 | 23 | 24 | 16 | 17 | 27 | 35.75 | 31.00 | 27.50 |
| Gemini 2.5 pro | 51 | 44 | 40 | 97 | 74 | **75** | 31 | 26 | 29 | 26 | 24 | 23 | 51.25 | 42.00 | 41.75 |
| Claude 3.7 Sonnet | 1 | 0 | 0 | 86 | 56 | 49 | **60** | 19 | 24 | **40** | 21 | 27 | 44.25 | 24.00 | 25.00 |
| Qwen VL 2.5 7B | 0 | 0 | 0 | 66 | 36 | 35 | 27 | 18 | 24 | 16 | 22 | 18 | 27.25 | 19.00 | 19.25 |
| Monet | 0 | 1 | 0 | 62 | 35 | 36 | 33 | 23 | 27 | 19 | 27 | 26 | 28.50 | 21.50 | 22.25 |
| CapImagine | 1 | 0 | 0 | 68 | 44 | 32 | 37 | 23 | 21 | 33 | 29 | 28 | 34.75 | 24.00 | 20.25 |
| Human | 100.00 | 100.00 | 100.00 | 98.86 | 94.00 | 95.20 | 100.00 | 91.33 | 89.33 | 100.00 | 54.40 | 52.33 | 99.72 | 84.93 | 84.22 |
| Random Guess | 0.00 | 0.00 | 0.00 | 25.00 | 25.00 | 25.00 | 25.00 | 25.00 | 25.00 | 25.00 | 25.00 | 25.00 | 25.00 | 25.00 | 25.00 |

# D. Clarification on Our Position

We would like to clarify our primary positioning: this paper focuses on precisely diagnosing the fundamental failures of current LVR methods to lay the crucial foundation for future improvements. While we do not propose a complete fix

for latent reasoning in this work, our causal analysis on representation degeneration explicitly identifies the root cause. Resolving this intrinsic issue requires designing entirely new training paradigms to regularize the latent space, which constitutes highly valuable future work rather than a direct patch.

Moreover, We would like to clarify that our paper is not intended as a paradigm-level rejection of latent reasoning in general. Rather, the scope of **"NOT YET"** is to analyze why current representative LVR implementations have not yet realized strong causal latent reasoning. By establishing a systematic causal analysis framework and providing a strong, interpretable text-space baseline, we equip the community with both the diagnostic tools and the necessary baselines to guide the development of genuinely causally effective LVR methods.

## E. Limitations and Future Work

The limitations of this work are threefold. First, our proposed text-form approach introduces higher inference latency compared to latent-based methods due to the autoregressive decoding of longer sequences. Second, CapImagine serves primarily as a verification probe to demonstrate the causality gap in current latent paradigms rather than an optimal solution. We acknowledge that natural language is inherently limited in granularity compared to the theoretical information capacity of high-dimensional latent spaces. Consequently, how to rigorously construct a high-quality, causal reasoning chain within the latent space (Zhang et al., 2026; Zeng et al., 2025) remains an unsolved and challenging objective for future exploration.

