# OpenReview forum: "Imagination Helps Visual Reasoning, But Not Yet in Latent Space"
_ICML.cc/2026/Conference — ICML 2026 regular_

### Official Review · Reviewer_i4cX · 2026-03-04

**Soundness:** 3
**Presentation:** 3
**Significance:** 3
**Originality:** 3
**Overall Recommendation:** 4
**Confidence:** 4

**Summary:**

This paper examines whether latent tokens in latent visual reasoning truly function as causal mediators in MLLMs. Through systematic interventions and probing analyses, the authors find that latent tokens are largely insensitive to input changes and have minimal influence on answers, suggesting limited semantic and causal contribution. Motivated by these findings, they propose CapImagine, a text-space imagination approach that converts intermediate visual manipulations into explicit textual reasoning, achieving consistent improvements over latent-based methods across multiple benchmarks.

**Compliance With Llm Reviewing Policy:**

Affirmed.

**Final Justification:**

Most of my concerns have been addressed during the rebuttal. Considering the gap between the diagnosis of latent tokens and the solution based on explicit textual reasoning, I keep the original rating.

**Key Questions For Authors:**

1. Is the latent degeneration phenomenon consistent across different backbone sizes (e.g., 32B or larger)? Does scale mitigate collapse?
2. Why was Qwen3-VL used when constructing the training data, yet Qwen2.5-VL was chosen as the backbone model in the experiments instead of Qwen3-VL?

**Limitations:**

Yes

**Strengths And Weaknesses:**

### Strengths
1. This paper challenges a rapidly growing research direction (e.g., latent visual reasoning) and addresses a fundamental and under-explored question: do latent tokens actually function as meaningful causal mediators?
2. The use of Causal Mediation Analysis provides a principled and interpretable framework. The intervention-based design (e.g., do(X), do(Z)) is conceptually strong and aligns well with mechanistic interpretability research.
3. The analysis and findings are comprehensive and insightful.
4. The proposed method CapImagine achieves consistent improvements across multiple benchmarks

### Weaknesses
1. Although the analysis is thorough and insightful and the proposed method is simple and effective, there is a gap between the diagnosis and the solution. The paper shows that current latent tokens are "ineffective", but instead of improving or redesigning latent reasoning, it shifts to explicit text-based reasoning. This makes the contribution appear more as an argument for explicit reasoning rather than a direct attempt to address the weaknesses of latent reasoning itself.
2. The baseline comparison is not comprehensive enough. Since the proposed method relies on explicit text-based reasoning, it should be compared with stronger text-based baselines, such as Qwen3-VL-Thinking, standard CoT or other "think-with-images" style methods. Without these comparisons, it is unclear whether the improvements stem from the specific design or simply from adopting explicit reasoning.
3. The paper convincingly shows that latent tokens are ineffective in current designs, but does not deeply explain why degeneration occurs. A stronger theoretical analysis would enhance impact.
4. All the analysis and experiments are conducted on 7B–8B scale models, with no investigation of larger model sizes. It remains unclear whether the observed latent collapse and weak causal effects persist in stronger MLLMs, which limits the generality of the conclusions.

---

> ### Author Rebuttal · Authors · 2026-03-31
>
> We extend our appreciation for reviewer-i4cX for your constructive suggestions and acknowledgement of our paper. We hope our illustration below could address your concerns.
>
> ---
> ### **W1: Insufficient Real Contribution to the LVR paradigm.**
> We wish to clarify our primary positioning: this paper focuses on precisely diagnosing the fundamental failures of current LVR methods to lay the crucial foundation for future improvements. While we do not propose a complete fix for latent reasoning in this work, our causal analysis on representation degeneration explicitly identifies the root cause. Resolving this intrinsic issue requires designing entirely new training paradigms to regularize the latent space, which constitutes highly valuable future work rather than a direct patch. By **establishing a systematic causal analysis framework and providing a strong, interpretable text-space baseline**, we equip the community with both the diagnostic tools and the necessary baselines to guide the development of genuinely causally effective LVR methods.
>
> ---
> ### **W2: Stronger Text-space Reasoning Baselines.**
> We completely agree that including state-of-the-art text-space reasoning methods validates the effectiveness of our approach. Following your constructive suggestion, we have evaluated `CapImagine` against a suite of strong recent baselines, including PAPO, Vision-R1, R1-OneVision, and LLaVA-OneVision, across diverse benchmarks (V*, HR-Bench, MME-Realworld-Lite, and BLINK). As shown in the table below, `CapImagine` consistently exhibits superior performance. This robust performance across multiple datasets thoroughly validates the efficacy and necessity of our proposed text-space imagination framework.
>
> | Text Reasoning | V* | HR-Bench | MME-RWL | BLINK |
> | :--- | :--- | :--- | :--- | :--- |
> | PAPO | 36.1 | 68.1 | — | 52.7 |
> | Vision-R1 | 80.1 | 60.9 | 47.9 | 51.0 |
> | R1-Onevision | 84.2 | — | 35.1 | 50.1 |
> | LLaVa-OneVision | 75.4 | 61.4 | 48.5 | 53.0 |
> | `CapImagine` | 85.9 | 72.4 | 54.8 | 54.8 |
>
> ---
> ### **W3: Lack of Theoretical Analysis.**
> We sincerely thank the reviewer for this insightful feedback. We entirely agree that a deeper mechanistic explanation strengthens the impact of our empirical findings. We will provide the possible mechanistic explanation in our final version. Prior literature [4, 5] demonstrates that standard Transformer hidden states suffer from representation degeneration, **naturally clustering into a narrow, anisotropic cone**. In standard discrete CoT, lm_head combined with argmax/sampling introduces a strong non-linear operation that **pulls representations out of this narrow space**. Latent tokens, operating purely continuously, lack this projection mechanism. Consequently, they remain trapped within this anisotropic cone, mathematically explaining the extreme cosine similarity and "collapse" we empirically observed.
>
> ---
> ### **W4: Lack of Scaling to Larger Models.**
> Due to computational budget constraints, we were unable to conduct experiments on models of that scale at this time. It is worth noting that current LVR methods predominantly focus on 7B-scale models to ensure accessibility and fair comparison. We consider exploring the scaling behaviors of latent degeneration to be a crucial direction for future investigation.
>
> ---
> **Q1: Is the latent degeneration phenomenon consistent across different backbone sizes? Does scale mitigate collapse?**
>
> In LVR research area, researchers predominantly focuses on 7B scale models, so this question currently remains underexplored due to computational constraint. However, for latent reasoning in LLM, some works have started scaling latent reasoning on larger scale models (30-70B)[1,2,3]. This insightful question shall be discussed in the very near future.
>
> **Q2: Why was Qwen3-VL used when constructing the training data, yet Qwen2.5-VL was chosen as the backbone model in the experiments instead of Qwen3-VL?**
>
> We selected Qwen2.5-VL-7B as the backbone to ensure a fair and direct comparison with other baselines, which are uniformly finetuned on Qwen2.5-VL-7B. Meanwhile, utilizing a more advanced model (Qwen3-VL) for constructing training data is a common and effective practice to guarantee high-quality data curation. For instance, recent works like Monet similarly utilize state-of-the-art models such as DeepSeek-R1 and Gemini 2.5 Pro for dataset construction while evaluating on standard backbones.
>
> ---
> **We hope these clarifications fully address your concerns and further highlight the value of our work.**
>
> ---
> [1] Soft Thinking: Unlocking the Reasoning Potential of LLMs in Continuous Concept Space. NeurIPS 2025
>
> [2] Text Generation Beyond Discrete Token Sampling. NeurIPS 2025
>
> [3] LLMs are Single-threaded Reasoners: Demystifying the Working Mechanism of Soft Thinking. ICLR 2026
>
> [4] How Contextual are Contextualized Word Representations? EMNLP2 019
>
> [5] Representation Degeneration Problem in Natural Language Generation

---

> > ### Author Rebuttal · Reviewer_i4cX · 2026-04-03
> >
> > Most of my concerns have been addressed, and I currently have no other questions. I keep the rating.

---

### Official Review · Reviewer_F3ky · 2026-03-09

**Soundness:** 3
**Presentation:** 2
**Significance:** 3
**Originality:** 2
**Overall Recommendation:** 4
**Confidence:** 4

**Summary:**

This paper investigates the role of latent tokens in latent visual reasoning models through a causal mediation analysis framework. The authors formulate the reasoning process as a causal chain from the input to latent tokens, and finally to the output answer. Through systematic interventions, they show that perturbations to latent tokens have minimal impact on the final prediction, and that variations in the input induce only limited changes in the latent representations. They further conduct probing analyses suggesting that latent tokens encode limited task-relevant visual semantics.

Motivated by these findings, the authors propose CapImagine, a text-based reasoning approach that replaces latent reasoning tokens with explicit textual descriptions of intermediate visual reasoning steps. They finetune Qwen2.5-VL-7B on a rewritten and filtered version of the Monet-SFT dataset. Experimental results on perception-centric and visual reasoning benchmarks show improved performance compared to latent reasoning baselines, along with competitive inference efficiency.

**Compliance With Llm Reviewing Policy:**

Affirmed.

**Final Justification:**

The rebuttal addressed most of my concerns. The remaining concern is the limited novelty of the paper. However, since most of my concerns are addressed, I'm increasing my score to weak accept.

**Key Questions For Authors:**

1. How does CapImagine differ from existing text-based multimodal reasoning approaches that rely on explicit chain-of-thought or textual intermediate supervision?

2. Have the authors evaluated, or can they comment on, settings where latent visual reasoning may retain an advantage over text-based imagination (e.g., tasks requiring complex spatial transformations or mental simulation with limited textual decomposability)?

3. In the causal mediation analysis, could the perturbations to latent tokens (e.g., noise injection or replacement) be too severe, effectively pushing the model out of its training distribution and encouraging it to ignore the latent pathway altogether?

**Limitations:**

yes

**Strengths And Weaknesses:**

## Strengths

1. The causal mediation study of latent visual reasoning (LVR) is a strong and timely contribution. The systematic interventions on both \( X \rightarrow Z \) and \( Z \rightarrow Y \) provide compelling evidence that current LVR implementations may not meaningfully utilize latent tokens.

2. The paper includes multiple forms of analysis (intervention, similarity analysis, and probing), which consistently support the claim that latent tokens encode limited task-relevant information.

3. The use of an open backbone (Qwen2.5-VL-7B) and a clearly described data rewriting and filtering pipeline improves reproducibility.


## Weaknesses

1. CapImagine is essentially a text-based reasoning model trained with explicit textual imagination traces. While effective, the methodological contribution is incremental and primarily consists of data reformulation and supervised fine-tuning, rather than a fundamentally new modeling approach.

2. Since CapImagine operates as a text-based reasoning model, it should be evaluated against state-of-the-art text-driven multimodal reasoning systems, not only latent reasoning baselines. This makes it difficult to assess whether the gains stem from the removal of latent tokens or from standard chain-of-thought style supervision.

3. Several baselines (e.g., GPT-4o, Gemini 1.5 Pro, LLaVA-NeXT) are not explicitly reasoning models, and more recent reasoning-focused versions of these models have been available for some time. Comparing against older, non-reasoning variants weakens the evaluations and does not reflect the current state of the field.

4. Many evaluated benchmarks admit valid textual reasoning traces. It remains unclear whether the proposed approach would retain its advantage on tasks that require complex spatial transformation or mental imagery with limited textual decomposition. Benchmarks such as *STARE* [1], *Hyperphantasia* [2], and the Spatial Relations and IQ-test splits of *BLINK* [3] specifically target mental simulation and spatial reasoning capabilities. Evaluating on such benchmarks and repeating the causal mediation analysis in these settings would provide a stronger test of the central claim regarding the ineffectiveness of latent reasoning.

5. Some figures are difficult to interpret. Figure 2 is visually dense and lacks sufficient explanation, making it hard to follow the key conclusions. In Figure 5, the y-axis is unclear.

## References

[1] L. Li, M. Bigverdi, J. Gu, Z. Ma, Y. Yang, Z. Li, Y. Choi, and R. Krishna,
"Unfolding spatial cognition: Evaluating multimodal models on visual simulations,"
arXiv preprint arXiv:2506.04633, 2025.

[2] M. S. Sepehri, B. Tinaz, Z. Fabian, and M. Soltanolkotabi,
"Hyperphantasia: A benchmark for evaluating the mental visualization capabilities of multimodal LLMs,"
in Proc. NeurIPS Datasets and Benchmarks Track, 2025.

[3] X. Fu, Y. Hu, B. Li, Y. Feng, H. Wang, X. Lin, D. Roth, N. A. Smith, W.-C. Ma, and R. Krishna,
"BLINK: Multimodal large language models can see but not perceive,"
in Proc. ECCV, 2024, pp. 148–166.

---

> ### Author Rebuttal · Authors · 2026-03-31
>
> We wholeheartedly thank the reviewer for the constructive comments. We respond to the main concerns as follows.
>
> ---
> ### **W1: Incremental Technical Contribution.**
> We respectfully wish to clarify the primary positioning and core contribution of our paper. As noted by Reviewer eJHe, *"CapImagine offers an elegantly simple, text-based approach...provides a strong and interpretable baseline for future reasoning tasks."* Our primary objective is not necessarily to introduce a highly complex new architecture, but rather to introduce a systematic causal analysis framework and establish a rigorous baseline for strict comparison with existing latent-based reasoning methods. **We aim to provide the community with a grounded perspective that demystifies current LVR methods and catalyzes more robust future research**.
>
> ---
> ### **W2&3: Latest Text-Space Reasoning Baselines.**
> Following the reviewer’s suggestion, we added comparisons with PAPO, Vision-R1, R1-OneVision, and LLaVA-OneVision. These additional models are meant as stronger text-reasoning references rather than fully controlled same-pipeline baselines. Our main controlled evidence still comes from the same-backbone, same-source-data comparison against Monet. Even under this broader comparison, CapImagine remains favorable, supporting that improvement is not simply from adopting any generic explicit reasoning format.
>
> | Table-1 | V* | HR-Bench | MME-RWL | BLINK |
> | :--- | :--- | :--- | :--- | :--- |
> | PAPO | 36.1 | 68.1 | — | 52.7 |
> | Vision-R1 | 80.1 | 60.9 | 47.9 | 51.0 |
> | R1-Onevision | 84.2 | — | 35.1 | 50.1 |
> | LLaVa-OneVision | 75.4 | 61.4 | 48.5 | 53.0 |
> | `CapImagine` | 85.9 | 72.4 | 54.8 | 54.8 |
>
> ---
> ### **W4: More Complex Visual Reasoning Benchmarks.**
> Following your advice, we have extensively tested `CapImagine` on a series of benchmarks (Table2-4). The experimental results reveal a consistent performance advantage for CapImagine over the Monet baseline, highlighting the generalizability of our text-space imagination method. We have also conducted causal analysis on several subset by setting all latent tokens as the same tensor denoted by $do(z)_1$. **To Ensure the pertubed latent tokens are in-distribution**, we replace the entire latent sequence with latent sequences generated under other input, which is denoted by $do(z)_2$. Consistent with the results from our paper, such pertubation yields negligible differences in the model response, indicating the minimal causal effect of latent tokens on the answer.
>
> | Table-2 Hyperphantasia | 7Seg-E | 7Seg-M | 7Seg-H | Dots-E | Dots-M | Dots-H | Linear-E | Linear-M | Linear-H | Para-E | Para-M | Para-H | AVG |
> | :--- | :--- | :--- | :--- | :--- | :--- | :--- | :--- | :--- | :--- | :--- | :--- | :--- | :--- |
> | Qwen2.5-VL | 0 | 0 | 0 | 66 | 36 | 35 | 27 | 18 | 24 | 16 | 22 | 18 | 21.83 |
> | Monet | 0 | 1 | 0 | 62 | 35 | 36 | 33 | 23 | 27 | 19 | 27 | 26 | 24.08 |
> | `CapImagine` | 1 | 0 | 0 | 68 | 44 | 32 | 37 | 23 | 21 | 33 | 29 | 28 | 26.33 |
>
> | Table-3 | BLINK - Spa | BLINK - IQ | STARE - 2D Trans | STARE - 3D Trans | STARE - Cube | STARE - Tangram | STARE - Temporal | STARE - Persp. | STARE - AVG |
> | :--- | :--- | :--- | :--- | :--- | :--- | :--- | :--- | :--- | :--- |
> | Qwen2.5-VL | 81.11 | 27.33 | 38.4 | 28.8 | 40.7 | 54.5 | 36.5 | 23.2 | 36.7 |
> | Monet | 78.33 | 28.67 | 32.4 | 26.5 | 55.4 | 48.1 | 32.1 | 27.2 | 35.3 |
> | `CapImagine` | 85.31 | 28.67 | 43.2 | 35.8 | 56.5 | 49.1 | 35.9 | 25.2 | 40.7 |
>
> | Table-4 Pertubation | Dots-E | Linear-E | Para-E | BLINK-Spa |
> | :--- | :--- | :--- | :--- | :--- |
> | Monet | 62 | 33 | 19 | 78.33 |
> | w/ $do(z)_1$ | 61 | 29 | 21 | 81.82 |
> | w/ $do(z)_2$ | 61 | 31 | 24 | 78.33 |
>
> ### **W5: Presentation Clarity**
> Thanks for your advice! We will rigorously improve the figures for better clarity in the revision.
>
> ---
> **Q1: Difference from text reasoning methods.**
> `CapImagine` is not generic CoT supervision. Its intermediate text is derived from concrete visual manipulations and the resulting visual state changes in interleaved data, rather than from free-form textual rationales alone. In this sense, the reasoning chain is grounded in explicit intermediate visual evidence. We will make this distinction clear in the paper.
>
> **Q2: Advantageous scenarios for LVR over text reasoning.**
> LVR provides a less constrained representational space, thus excelling in tasks requiring complex **visual imagination** and **spatial intelligence**. For modeling spatial perception or simulating viewpoint changes, latent tokens are generally more expressive than explicit words for capturing fine-grained spatial details [1].
>
> **Q3: Could Pertubation be too severe?**
> This insightful point was also raised by Reviewer TUed. Due to length constraints, we  could only kindly refer you to our detailed discussion in the response to Reviewer TUed W2.
>
> We sincerely hope we have addressed your concerns!
>
> ---
> [1] Visual Generation Unlocks Human-Like Reasoning through Multimodal World Models

---

> > ### Author Rebuttal · Reviewer_F3ky · 2026-04-03
> >
> > I appreciate the authors’ efforts in addressing several of my concerns. In particular, the addition of stronger text-based baselines and evaluations on more challenging benchmarks significantly strengthens the empirical section and provides better support for the claims.
> >
> > I also acknowledge that the primary contribution of the paper lies in its grounded causal perspective on LVR methods, which offers useful insights into their limitations. However, I still find the overall methodological novelty to be limited.
> >
> > Overall, I believe the paper presents a valuable diagnostic study with improved experimental validation after rebuttal, and I am willing to update my score to 4 (weak accept).

---

> > > ### Author Response · Authors · 2026-04-06
> > >
> > > Dear Reviewer F3ky:
> > >
> > > Thank you very much for your continuous engagement and for acknowledging our efforts during the rebuttal phase. We are thrilled to hear that the stronger text-based baselines and additional experiments addressed your empirical concerns, and that you find our diagnostic study and causal perspective on LVR methods valuable.
> > >
> > > We deeply appreciate your willingness to raise your score to a 4 (weak accept). However, we would like to bring a gentle and kind reminder to the dearest reviewer, as it appears **the score has not yet been officially updated in the openreview system**. We would **highly and sincerely appreciate it** if you could officially update your score accordingly!
> > >
> > > Thank you once again for your constructive feedback and guidance, which have genuinely helped improve our work.

---

### Official Review · Reviewer_TUed · 2026-03-11

**Soundness:** 3
**Presentation:** 3
**Significance:** 3
**Originality:** 3
**Overall Recommendation:** 4
**Confidence:** 4

**Summary:**

At a time when the research community is very excited about latent visual reasoning in vision-language models, this paper takes a calm and careful step back to ask a simple but important question: do the latent visual tokens that these models generate actually do anything useful? The paper studies this through two types of intervention experiments. First, it replaces the visual latent variables with random noise (input intervention). Second, it directly perturbs the intermediate latent tokens during generation (latent variable intervention). Both experiments point to the same conclusion: the latent variables produced by current models like Monet are highly homogeneous across different questions, and disturbing them barely changes the final answer.

**Compliance With Llm Reviewing Policy:**

Affirmed.

**Key Questions For Authors:**

Please refer to the "Weaknesses" part.

**Limitations:**

Yes

**Strengths And Weaknesses:**

Strengths:

(1) The intervention experiments are well-designed and the conclusion is convincing.

(2) CapImagine makes a strong practical point with a very simple design. The fact that a text-description-based method trained on 17k clean samples beats a complex latent-space model trained on 125k noisy samples is a strong and clear result.

(3) The paper is well-written and easy to follow. The motivation, diagnosis, and proposed solution are connected in a logical and clear way.

Weaknesses:

(1) The paper demonstrates two things: (a) CapImagine with 17k clean data beats Monet with 125k noisy data, and (b) CapImagine with 17k clean data beats CapImagine with 125k noisy data. From these two results, the paper argues that latent space reasoning is flawed and that text-based description is the right approach. However, there is a simpler explanation that the paper does not rule out: the performance gap might mainly come from data quality differences rather than the modeling approach itself. The missing experiment is straightforward: train Monet on the same 17k clean data and compare it with CapImagine trained on the same data. If Monet, with 17k clean data, still falls significantly behind CapImagine, with 17k clean data, then the paper's conclusion about the problem with latent reasoning is strongly supported.

(2) When the paper injects random noise to replace latent tokens and observes that the model output barely changes, the interpretation is that the latent tokens are not causally important. However, there is a more tedious explanation: modern language models are trained to be robust and tend to ignore tokens that appear out of distribution. When the model sees random noise tokens in a position where it expects a structured latent visual representation, it may simply treat those tokens as anomalous and fall back on the surrounding image and text tokens to produce an answer based on prior knowledge, rather than ever engaging with the latent tokens.

(3) The category-level analysis is incomplete, and it is unclear whether CapImagine truly helps with spatial and non-textualizable visual reasoning. The paper should provide more detailed results by question category across all benchmarks, not just V*Bench. Specifically, it would be very useful to see whether CapImagine gains are concentrated in language-friendly categories (attributes, object recognition) and flat or even negative in language-unfriendly categories (spatial reasoning, counting, geometric reasoning). If that is the case, it would suggest that CapImagine is not a general solution to visual reasoning but rather a good solution for a specific subset of tasks where language description is sufficient.

---

> ### Author Rebuttal · Authors · 2026-03-31
>
> We sincerely thank the reviewer for the concrete suggestions. Following them, we added new controlled experiments and broader evaluations.
>
> ---
> ### **W1: Training Monet with the Same 17k Data.**
> We reproduced the **entire training procedure** for Monet using the same 17k clean data applied in CapImagine. This experiment was conducted on eight A800 (80GB) GPUs and took approximately 24 hours. As demonstrated in Table-1, training Monet with this reduced dataset yields only a marginal performance boost, falling significantly behind CapImagine. As shown below, Monet-17k remains clearly below CapImagine, indicating that the performance gap cannot be explained by data quality alone.
>
> **Table-1 Evaluation of Monet-17k**
> | Model | V* (Overall) | V* (Direct) | V* (Spatial) | HRBench-4K (Overall) | HRBench-4K (FSP) | HRBench-4K (FCP) | HRBench-8K (Overall) | HRBench-8K (FSP) | HRBench-8K (FCP) |
> | :--- | :--- | :--- | :--- | :--- | :--- | :--- | :--- | :--- | :--- |
> | Monet-17k | 79.6 | 82.6 | 75.0 | 70.7 | 88.0 | 53.3 | 67.9 | 84.0 | 51.8 |
> | Monet | 83.3 | 83.5 | 82.9 | 71.0 | 85.3 | 56.8 | 68.0 | 79.8 | 56.3 |
> | `CapImagine` | 85.9 | 87.8 | 82.9 | 74.1 | 88.5 | 59.8 | 70.7 | 84.8 | 56.5 |
>
> ---
> ### **W2: Injecting Gaussian Noise Yields OOD Issue.**
> We would like to address this valid concern from two distinct perspectives:
> - First, in practice, we calculate the mean and standard deviation of the target latent token before creating Gaussian noise with strictly identical statistics. Through this design, we constrain the noise to have the **exact same intensity** as the original embedding, largely ensuring that the modified latent token remains in-distribution.
> - Second, even if we assume the injected Gaussian noise pushes the latent token to OOD, causing the model to ignore the latent reasoning process, **our conclusion still holds**. Under this hypothetical setting, the model reasons directly from the input to the final answer ($X \rightarrow Y_1$), circumventing and ignoring the latent reasoning chain. We can then form a strict comparison between the original latent reasoning process ($X \rightarrow Z \rightarrow Y_0$) and our perturbed case ($X \rightarrow Y_1$). The identicalness between $Y_0$ and $Y_1$ actually demonstrates that the latent token $Z$ is not causally important to the final output. Therefore, even in an OOD scenario, the empirical results strongly support our claim.
>
> ---
> ### **W3: Broader Evaluation on Complex Visual and Imagination-Intensive Benchmarks.**
> Following the reviewer’s suggestion, we additionally evaluate on STARE, Hyperphantasia, and BLINK subsets targeting more imagination-intensive and spatial reasoning settings. The experimental results reveal a consistent performance advantage for CapImagine over the Monet baseline, highlighting the generalizability of our text-space imagination method not only in language-friendly tasks but more complex scenarios such as spatial perception and counting.
>
> **Table-2: Evaluation on Hyperphantasia**
> | Model | 7Seg(E) | 7Seg(M) | 7Seg(H) | Dots(E) | Dots(M) | Dots(H) | Linear(E) | Linear(M) | Linear(H) | Parabolic(E) | Parabolic(M) | Parabolic(H) | AVG |
> | :--- | :--- | :--- | :--- | :--- | :--- | :--- | :--- | :--- | :--- | :--- | :--- | :--- | :--- |
> | Qwen2.5-VL | 0 | 0 | 0 | 66 | 36 | 35 | 27 | 18 | 24 | 16 | 22 | 18 | 21.83 |
> | Monet | 0 | **1** | 0 | 62 | 35 | **36** | 33 | 23 | **27** | 19 | 27 | 26 | 24.08 |
> | `CapImagine` | **1** | 0 | 0 | **68** | **44** | 32 | **37** | **23** | 21 | **33** | **29** | **28** | **26.33** |
>
> **Table-3: Evaluation on BLINK subset and STARE**
> | Model | BLINK - Spa | BLINK - IQ | STARE - 2D Trans | STARE - 3D Trans | STARE - Cube | STARE - Tangram | STARE - Temporal | STARE - Persp. | STARE - AVG |
> | :--- | :--- | :--- | :--- | :--- | :--- | :--- | :--- | :--- | :--- |
> | Qwen2.5-VL | 81.11 | 27.33 | 38.4 | 28.8 | 40.7 | 54.5 | 36.5 | 23.2 | 36.7 |
> | Monet | 78.33 | 28.67 | 32.4 | 26.5 | 55.4 | 48.1 | 32.1 | 27.2 | 35.3 |
> | `CapImagine` | 85.31 | 28.67 | 43.2 | 35.8 | 56.5 | 49.1 | 35.9 | 25.2 | 40.7 |

---

> > ### Author Rebuttal · Reviewer_TUed · 2026-04-04
> >
> > Thanks for the detailed response and additional discussions. Most of my concerns have been addressed, and I currently have no more questions. I will maintain my positive score.

---

### Official Review · Reviewer_eJHe · 2026-03-13

**Soundness:** 2
**Presentation:** 2
**Significance:** 2
**Originality:** 3
**Overall Recommendation:** 3
**Confidence:** 4

**Summary:**

This paper investigates the internal mechanisms of latent visual reasoning in MLLMs using causal mediation analysis to determine if latent tokens genuinely perform deliberative reasoning. The authors find that latent tokens in existing models exhibit high homogeneity and have minimal causal impact on the final prediction, suggesting they function more like static soft prompts than active carriers of visual semantics. To address this limitation, the paper introduces CapImagine, an alternative method that replaces latent-space reasoning with explicit text-based imagination by generating textual descriptions of intermediate visual manipulations. Experimental evaluations demonstrate that this straightforward text-space approach consistently outperforms several complex latent-space baselines across multiple vision-centric benchmarks.

**Compliance With Llm Reviewing Policy:**

Affirmed.

**Final Justification:**

I thank the authors for their response and clarification. I appreciate their commitment to revising the title, abstract, and conclusion. This will ensure the paper does not read as a broad rejection of latent reasoning. However, my assessment must rely on the current manuscript.

My core concern remains. The draft extrapolates broad conclusions from baselines with shared, confounding design choices (e.g., exposure bias, representation mismatch). The authors acknowledge these limitations. Yet, reframing the narrative to diagnose these specific failures requires a substantial revision. The focus must shift away from critiquing latent reasoning entirely.

These structural changes are necessary. I cannot fully evaluate them without a revised draft. Therefore, I maintain my initial score of Weak Reject.

**Key Questions For Authors:**

See weakness.

**Limitations:**

yes

**Strengths And Weaknesses:**

### Strengths

* **Tackles a Critical Blind Spot:** The paper investigates an important, under-explored assumption in MLLMs—whether latent space is actually performing deliberative reasoning or just acting as a passthrough.
* **Valuable Conceptual Insight:** The observation that latent tokens essentially function as "soft prompts" or placeholders, rather than active carriers of visual imagination, is a highly intuitive and valuable contribution to the field.
* **Simple and Effective Baseline:** The proposed alternative, `CapImagine`, offers an elegantly simple, text-based approach to visual imagination that provides a strong and interpretable baseline for future reasoning tasks.


### Weaknesses

* **Information Bottleneck and Lossy Compression Confound Core Conclusions:** The paper's sweeping conclusion—that latent visual reasoning is inherently flawed—is not rigorously supported due to a severe information bottleneck in the experimental design. The evaluated latent baselines (Mirage, Monet, LVR) restrict the latent tokens to extremely small sizes ( $k<=16$). Furthermore, the mechanisms used to achieve this compression are highly lossy: Mirage relies on average pooling, while LVR relies on sparse index selection. The observed token homogeneity and "soft prompt" behavior are the expected mathematical artifacts of forcing high-dimensional visual data through this aggressive dimensionality reduction, rather than proof of a fundamental failure in latent-space reasoning itself.
* **Unfair Baseline Comparison:** By comparing these heavily compressed, lossy latent states against fully explicit text-space reasoning (CapImagine), it sets up an uneven baseline comparison. Text-based captions inherently utilize a much larger token budget to describe visual transformations. It has effectively demonstrated that *highly compressed* bottlenecks fail at visual reasoning, but they have not proven their broader, title-level claim against latent-space imagination as a whole.

---

> ### Author Rebuttal · Authors · 2026-03-31
>
> We extend sincere appreciation for reviewer-eJHe, for your insightful perception into the field of Latent Visual Reasoning (LVR) research.  Here, we would like to address your concern through the following Table and three different aspects.
>
> ---
> ### **W1: Information Bottleneck.**
> While it may be easily overlooked, existing LVR methods have **already explored** extending the length of latent tokens. As shown in the table above, LVR directly adopts **visual tokens from Region-of-Interest (ROI) images** and trains the model to conduct latent reasoning on an average of **91.78*** visual tokens, where the model could sufficiently express latent thought and visually-related semantics. The latent size is only set to a fixed number during inference due to decoding collapse issues. Similarly, Monet explores extending the latent reasoning length to **100 tokens** during inference (Figure 5 of Monet), yet yielding only a marginal performance improvement. Therefore, the perceived information bottleneck in fact hardly exists than assumed.
>
> **Table-1: Comparison of the supervision signal and performance among different methods at various reasoning length**
> | Method | Supervision | Reasoning length | Performance (on V*) |
> | :--- | :--- | :--- | :--- |
> | LVR(K=8) | ROI ViT Feature | 8 | 81.7 |
> | LVR(K=100)* | ROI ViT Feature | 100 | 81.6 |
> | Monet(K=16) | Distillation | 12 | 82.3 |
> | Monet(K=100) | Distillation | 100 | 81.7 |
> | `CapImagine` | Text Label | 49.1 | 85.9 |
>
> ### **W2: Unfair Baseline Comparison.**
> On average, CapImagine conducts the visual imagination process using approximately **49.1 tokens** as shown in Table-1, which falls into the same order of magnitude as other LVR methods. By comparison, CapImagine utilizes inference computation more efficiently and achieves superior performance than latent-based methods across various latent sizes.
>
> ### **W3: Lossy Compression.**
> Although the average pooling method used in Mirage does introduce information loss, LVR and Monet are largely unaffected by lossy compression. LVR utilizes all visual tokens from the ROI image, forcing the model to output uncompressed visual semantics. Conversely, Monet optimizes teacher representations sourced from both visual semantics and text information, rather than relying directly on compression. Overall, their supervision signals for latent tokens **do not suffer from severe information loss brought by compression**. These methods have rigorously and comprehensively explored the potential of latent-based reasoning methodologies.
>
> ---
> Together, we wholeheartedly hope that we have addressed your concerns and elevated your estimation of our paper! If you have any follow-up questions, we are more than delighted to further discuss with you.
>
> ---
> *By setting latent size to 100, LVR matches the lengthy latent tokens during its training process and consequently largely eliminates the information bottleneck issue.
> *We implement statistical analysis on the quantity of encoded visual tokens for the randomly sample 20k ROI images in the training dataset.

---

> > ### Author Rebuttal · Reviewer_eJHe · 2026-04-04
> >
> > Thank you to the authors for their detailed rebuttal and the additional experimental results provided. However, after carefully reviewing the response, my core concerns regarding the paper's sweeping claims and the validity of the baseline comparisons remain largely unaddressed.
> >
> > **1. Unsubstantiated Claims and Train/Test Mismatch Regarding Information Bottlenecks**
> > The authors argue that existing methods have already explored extending latent token lengths, citing LVR training on an average of 91.78 tokens. I find these arguments unconvincing for two reasons:
> > * **Lack of Evidence for LVR:** I cannot find evidence in the existing literature to support the claim that LVR trains on an average of 91.78 visual tokens.
> > * **Severe Distribution Shift in Monet:** As explicitly stated in Figure 5 of the Monet paper, the model's training latent size is limited to K ∈ [8, 10, 12]. Evaluating a model at an inference size of K=100 when it was only trained on K ≤ 12 introduces a massive train-test distribution mismatch. The lack of performance improvement at K=100 is entirely expected due to this discrepancy, and therefore cannot be used as proof that "the perceived information bottleneck in fact hardly exists."
> >
> > **2. Misunderstanding of "Lossy Compression"**
> > The authors' response in W3 indicates a misunderstanding of my original critique regarding lossy compression. The rebuttal defends the supervision signals regarding local ground-truth regions. However, my point is that the compression is highly lossy relative to the **native representation of the MLLM's vision encoder**, which processes the image globally. By forcing the reasoning process into these restricted, non-native local representations, the baselines discard the rich, global visual context the MLLM natively relies on.
> >
> > **3. Core Concern: Extrapolating Broad Claims from Specific Implementations**
> > My primary concern remains the title-level claim that latent visual reasoning is inherently limited. While the paper effectively demonstrates that the evaluated baseline methods underperform compared to text-space reasoning (CapImagine), extrapolating this to critique the broader concept of latent-space reasoning lacks methodological rigor.
> >
> > The baselines analyzed are not orthogonal implementations of latent reasoning; rather, they share specific, coupled design choices that act as confounding variables in your evaluation. Specifically:
> > * They rely on pooled embeddings, resulting in lossy compression relative to native MLLM representations.
> > * Their use of selected tokens is not natively aligned with the MLLM pretrained space.
> > * They introduce potential exposure bias (e.g., projecting output hidden states to align with visual tokens during training creates a mismatch when those generated tokens must serve as autoregressive context during inference).
> >
> > Because the evaluated methods share these specific design choices, it is difficult to isolate whether the performance drop is due to a fundamental limitation of "latent visual reasoning" itself, or simply the natural consequence of these shared constraints.
> >
> > I strongly encourage the authors to tone down the broad generalizations and reframe the paper around investigating the *root causes* of why these specific implementations fail (i.e., the impact of extreme bottlenecks, non-native representations, and exposure bias).
> >
> > Until the claims are properly scoped to match the limitations of the evaluated methods, I maintain my current score.

---

> > > ### Author Response · Authors · 2026-04-06
> > >
> > > We sincerely thank Reviewer eJHe for the careful follow-up.
> > >
> > > We would like to clarify that our paper is not intended as a paradigm-level rejection of latent reasoning in general. Rather, **the scope of "not yet"** is to analyze why current representative LVR implementations have not yet realized strong causal latent reasoning. To avoid broader interpretations, we will make this scope more explicit in the revised title/abstract/conclusion.
> > >
> > > Regarding the LVR supervision signal, the 91.78-token statistic is an empirical measurement from the official LVR data pipeline: the supervision targets are ROI-aligned ViT tokens selected from the original visual token sequence. These tokens are extracted after full-image visual encoding, and therefore are not literal isolated local crops; they already contain globally contextualized information through self-attention. Our point is not that this removes all concerns about representation mismatch, but that the supervision signal in LVR is more faithful to the native visual representation than a purely local-crop interpretation would suggest.
> > >
> > > We also agree that Monet evaluated at K=100 should not be over-interpreted because of the train/test mismatch beyond its training range, and we will revise the wording accordingly. More broadly, we acknowledge that current LVR baselines may share coupled design choices, including possible representation restrictions and train/inference mismatch. Our contribution is therefore to diagnose why these prevailing implementations, despite their promise, have **not yet** shown strong causal dependence on latent mediators, rather than to claim that latent-space reasoning itself is inherently limited or impossible.
> > >
> > > We hope this clarification better reflects both the evidence and the intended scope of our paper.

---

### Decision · Program_Chairs · 2026-04-30

**Decision:**

Accept (regular)

**Comment:**

This paper presents a timely and rigorous causal mediation analysis of Latent Visual Reasoning (LVR) in MLLMs. The authors compellingly demonstrate that latent tokens in current models exhibit high homogeneity and exert minimal causal influence on predictions, acting more like static placeholders than active reasoning agents. To address this, the authors introduce a strong, interpretable text-based baseline (CapImagine) that outperforms complex latent-space methods.

The reviewers praised the paper for tackling a critical blind spot in LVR. During the rebuttal, the authors successfully resolved the committee's primary concerns by:

Controlling for Data Quality: Training the Monet baseline on the same 17k clean dataset as CapImagine, confirming the performance gap is not merely an artifact of training data differences.

Expanding Evaluations: Adding comparisons against stronger text-based reasoning baselines (e.g., Vision-R1) and more complex spatial/imagination benchmarks (Hyperphantasia, BLINK, STARE).

Providing Theoretical Grounding: Offering a plausible mechanistic explanation for the observed latent degeneration (continuous hidden states clustering in an anisotropic cone).

While one reviewer maintained a weak reject score due to concerns that the evaluated baselines suffer from confounding issues like lossy compression and exposure bias, the consensus is that diagnosing these exact flaws in prevailing LVR methods is a highly valuable contribution to the community.